# Infection of Brain Pericytes Underlying Neuropathology of COVID-19 Patients

**DOI:** 10.3390/ijms222111622

**Published:** 2021-10-27

**Authors:** Matteo Bocci, Clara Oudenaarden, Xavier Sàenz-Sardà, Joel Simrén, Arvid Edén, Jonas Sjölund, Christina Möller, Magnus Gisslén, Henrik Zetterberg, Elisabet Englund, Kristian Pietras

**Affiliations:** 1Department of Laboratory Medicine, Division of Translational Cancer Research, Lund University Cancer Centre, Medicon Village, Lund University, 223 81 Lund, Sweden; matteo.bocci@med.lu.se (M.B.); clara.oudenaarden@med.lu.se (C.O.); jonas.sjolund@med.lu.se (J.S.); christina.moller@med.lu.se (C.M.); 2Department of Clinical Sciences Lund, Division of Pathology, Lund University, 221 84 Lund, Sweden; Xavier.SaenzSarda@skane.se (X.S.-S.); elisabet.englund@med.lu.se (E.E.); 3Department of Psychiatry and Neurochemistry, Institute of Neuroscience & Physiology, The Sahlgrenska Academy at the University of Gothenburg, 431 41 Gothenburg, Sweden; joelsimren1@gmail.com (J.S.); henrik.zetterberg@clinchem.gu.se (H.Z.); 4Department of Infectious Diseases, Institute of Biomedicine, The Sahlgrenska Academy at the University of Gothenburg, 413 46 Gothenburg, Sweden; arvideden@gmail.com (A.E.); magnus.gisslen@infect.gu.se (M.G.); 5Region Västra Götaland, Department of Infectious Diseases, Sahlgrenska University Hospital, 416 50 Gothenburg, Sweden; 6Clinical Neurochemistry Laboratory, Sahlgrenska University Hospital, 431 80 Gothenburg, Sweden; 7Department of Neurodegenerative Disease, UCL Institute of Neurology, London WC1N 3BG, UK; 8UK Dementia Research Institute at UCL, London WC1E 6BT, UK

**Keywords:** COVID-19, pericytes, ACE2, SARS-CoV-2, blood–brain barrier, multiplexed IHC, brain, vasculature, infection

## Abstract

A wide range of neurological manifestations have been associated with the development of COVID-19 following SARS-CoV-2 infection. However, the etiology of the neurological symptomatology is still largely unexplored. Here, we used state-of-the-art multiplexed immunostaining of human brains (*n* = 6 COVID-19, median age = 69.5 years; *n* = 7 control, median age = 68 years) and demonstrated that expression of the SARS-CoV-2 receptor ACE2 is restricted to a subset of neurovascular pericytes. Strikingly, neurological symptoms were exclusive to, and ubiquitous in, patients that exhibited moderate to high ACE2 expression in perivascular cells. Viral dsRNA was identified in the vascular wall and paralleled by perivascular inflammation, as signified by T cell and macrophage infiltration. Furthermore, fibrinogen leakage indicated compromised integrity of the blood–brain barrier. Notably, cerebrospinal fluid from additional 16 individuals (*n* = 8 COVID-19, median age = 67 years; *n* = 8 control, median age = 69.5 years) exhibited significantly lower levels of the pericyte marker PDGFRβ in SARS-CoV-2-infected cases, indicative of disrupted pericyte homeostasis. We conclude that pericyte infection by SARS-CoV-2 underlies virus entry into the privileged central nervous system space, as well as neurological symptomatology due to perivascular inflammation and a locally compromised blood–brain barrier.

## 1. Introduction

The clinical manifestations of coronavirus disease 2019 (COVID-19) infection primarily include respiratory symptoms, ranging from a mild cough to severe bilateral pneumonia [1,2]. However, SARS-CoV-2 bears an organotropism beyond the respiratory tract [3,4], with increasing testimony indicating the brain as an extrapulmonary target of SARS-CoV-2 [5]. The involvement of the central nervous system (CNS) encompasses a broad spectrum of neurological manifestations (including headache, fatigue, anosmia, ageusia, confusion, and loss of consciousness), often representing an ulterior clinical morbidity that significantly contributes to COVID-19-related deaths [6,7,8].

The main entry receptor for SARS-CoV-2 is reported to be the angiotensin-converting enzyme 2 (ACE2), which is a component of the renin–angiotensin system [9,10]. To date, there is still no conclusive evidence concerning the localization of ACE2 in the human CNS [3], and the mechanism of SARS-CoV-2 infection in the brain remains a conundrum.

Here, using highly sensitive multiplexed immunohistochemistry (mIHC) of brain tissue from a series of confirmed COVID-19 patients and corresponding controls, we determined that ACE2 is exclusively expressed by brain pericytes in the subset of patients that also exhibited neurological symptoms. Moreover, spatial immunophenotyping revealed a localized perivascular inflammation in brain tissue from COVID-19 patients, paralleled by an impairment of the functionality of the vascular wall as indicated by loss of integrity of the blood–brain barrier (BBB). Finally, in the cerebrospinal fluid (CSF) of a cohort of COVID-19 patients with neurological involvement, levels of soluble PDGFRβ, a pericyte-specific marker in the brain, were significantly reduced compared with non-COVID-19 individuals, suggestive of SARS-CoV-2-related functional impairment of pericytes. Taken together, our findings highlight a previously unappreciated role for brain pericytes in acting as pioneers for SARS-CoV-2 entry into the CNS. 

## 2. Results

### 2.1. The ACE2 Receptor Is Expressed by Pericytes in Murine and Human Brains

Expression of ACE2 in the brain has variably been reported in neurons, glial cells including astrocytes, and vascular cells [11,12,13,14,15]. Because of this ambiguity of localization, we started by exploring ACE2 expression in publicly available mRNA and protein datasets from murine and human brains. Mining of the Allen Mouse Brain Atlas of single-cell transcriptomes demonstrated unique enrichment for *Ace2* transcript in pericytes (Figure 1A). A similar compartmentalization was observed in the *Tabula Muris* [16] and in a single-cell RNA sequencing (scRNA-seq) compendium of the murine brain vasculature [17,18] (Appendix A). In agreement with the transcriptional data, localization of the ACE2 protein by the Human Protein Atlas [19] was restricted to the perivascular compartment in a subset of blood vessels in the human cerebral cortex (Appendix A). 

### 2.2. The ACE2 Protein Is Expressed by Perivascular Cells of Neural Tissue from COVID-19 Patients with Neurological Symptoms

Next, we sought to investigate the expression of ACE2 in the brain tissue of COVID-19 patients. To this end, we obtained FFPE samples of multiple brain regions from six patients whose death was confirmed to be a consequence of SARS-CoV-2 infection and from seven control cases (Appendix A). In the frontal cortex, moderate to high ACE2 immunoreactivity revealed a vascular pattern in a subset of blood vessels in 5 of the 13 cases (Figure 1B). Reassuringly, other brain regions showed an equivalent distribution of ACE2, indicating that ACE2 was widely expressed in perivascular cells throughout the CNS (Figure 1C). Notably, ACE2 reactivity, which was confirmed with two different antibodies in positive control tissues from the kidney (Appendix A), appeared to be a patient-specific feature, since some cases did not show positivity at all, or showed signals with very low frequency (Figure 1D and Appendix A). To conclusively validate which cell type harbored ACE2 expression, we performed mIHC on human brain tissue to simultaneously visualize ACE2, CD31^+^ endothelial cells, and PDGFRβ^+^ pericytes. ACE2 expression coincided with that of PDGFRβ, but not with CD31 staining (Figure 1E and Appendix A). Pericytes investing the vasculature exhibited a nuanced pattern of PDGFRβ and ACE2 immunoreactivity, with some cells bearing positivity solely for PDGFRβ, while other perivascular cells simultaneously expressed both PDGFRβ and ACE2 markers. Remarkably, the three COVID-19 patients that exhibited moderate to high perivascular ACE2 expression in the brain all presented with neurological symptoms, while all ACE2-negative patients remained free from such manifestations (Figure 1D). Collectively, our data demonstrate that in the brain, ACE2 is exclusively expressed by pericytes in a manner that signifies the development of neurological symptoms from COVID-19.

### 2.3. SARS-CoV-2 Is Detectable in the Human Brain of COVID-19 Patients

An increasing body of evidence converges on the inherent difficulty of detecting SARS-CoV-2 in the brain [20,21]. To build on previous reports on the localization of SARS-CoV-2 in human brain tissue, we additionally analyzed brain samples from noninfected individuals to enable conclusions about the presence of the spike protein or the nucleocapsid protein of SARS-CoV/SARS-CoV-2 in the CNS with a higher certainty. For both viral components, positive areas in brain sections of COVID-19 patients exhibited patterns comparable with those shown in previous studies [22]. Notably, however, we demonstrated an analogous intensity and distribution of the viral proteins when we probed brain tissues from noninfected individuals (Figure 2A). 

In order to unequivocally define our ability to visualize viral particles in human tissues, we gained access to placental tissue from a confirmed case of SARS-CoV-2 vertical transmission to serve as a positive control [23]. We also made use of the J2 antibody specifically designed to detect viral double-stranded (ds)RNA. In the placenta, a 7-plex mIHC panel confirmed the epithelial cytokeratin^+^ syncytiotrophoblasts as the main target for viral infection by virtue of expression of ACE2 and the presence of dsRNA in a well-defined dotted pattern (Figure 2B and Appendix A), a pattern of distribution which was essentially preserved with antibodies against the *Coronaviridae* family or SARS-CoV-2-specific antigens (Appendix A). Finally, applying the now-validated protocol for detection of viral dsRNA to brain sections, we identified an analogous dotted pattern in discrete perivascular, non-endothelial, cells in the brain of COVID-19 patients (Figure 2C and Appendix A). Reassuringly, the perivascular staining pattern was absent from brain samples of noninfected individuals. Together with our observations of ACE2 expression in pericytes, our conclusive localization of viral dsRNA suggests that brain pericytes are indeed uniquely susceptible to viral infection and may serve as CNS entry points for SARS-CoV-2.

### 2.4. Perivascular Infection by SARS-CoV-2 in the Brain Is Paralleled by Perivascular Inflammation

We hypothesized that infection of pericytes would result in neuroinflammation and therefore implemented a spatial immunophenotyping approach for the concomitant detection of the endothelium (CD34^+^) and five immune cell populations, including T helper and cytotoxic T lymphocytes, regulatory T cells, B cells, and macrophages. Surrounding the brain vasculature in COVID-19 patients, we detected CD4^+^ and CD8^+^ T cells, as well as CD68^+^ macrophages, indicative of perivascular inflammation, rather than widespread neuroinflammation in the brain parenchyma (Figure 2D and Appendix A). The immune infiltration did not affect all blood vessels, indicating that the inflammation was not the result of systemic mediators, but rather of local instigation.

### 2.5. Pericyte Infection Leads to Vascular Fibrinogen Leakage in the CNS 

Next, we investigated whether impaired pericyte function subsequent to SARS-CoV-2 infection and the perivascular inflammation impinged on the integrity of the vascular wall. We first performed a 7-plex mIHC staining focusing on the permeability of the neurovascular unit. Remarkably, in COVID-19 patients, extravascular fibrinogen was readily detected as a characteristic gradient in subsets of vessels, occasionally also characterized by ACE2 expression and the presence of viral dsRNA (Figure 3A and Appendix A). Conversely, fibrinogen was fully retained within the blood vessels of noninfected control cases. Moreover, astrocyte priming indicative of local activation of the brain parenchyma was not apparent during COVID-19 infection (Figure 3B and Appendix A). Together with our identification of SARS-CoV-2 and immune cell infiltrates in the perivascular region, the leakage of fibrinogen from the blood vessels strongly suggests that viral infection of pericytes breaches the tightly organized BBB.

### 2.6. Shedding of PDGFRβ into the CSF Is Reduced in COVID-19 Patients

Our findings led us to speculate that the homeostatic state of brain pericytes would be disrupted in COVID-19 patients. Therefore, we collected CSF from an additional eight patients with acute COVID-19 that presented with neurological manifestations, as well as noninfected matched controls (Appendix A). Intriguingly, the soluble level of the pericyte marker sPDGFRβ in the CSF of COVID-19 patients was on average significantly lower than that in non-COVID-19 control individuals as measured by ELISA, indicative of a perturbed pericyte homeostasis (Figure 3C).

## 3. Discussion

The primary cellular receptor for SARS-CoV-2 entry is ACE2 [9], but the expression pattern of ACE2 in the CNS has not been conclusively resolved. Notably, the few published studies detailing the expression of ACE2 and/or SARS-CoV-2 protein in the CNS lack reliable and appropriate controls, precluding firm conclusions. Here, by means of highly sensitive mIHC and the use of both positive and negative control tissues, we were able to confirm that ACE2 exhibited an exclusive perivascular expression pattern in the CNS. Similarly, viral particles and their dsRNA were observed in CNS pericytes in COVID-19 patients, independently of the perivascular ACE2 expression status. Whether other coreceptors for SARS-CoV-2, including TMPRSS2, CD147, and neuropilin-1, contribute to CNS tropism remains to be investigated.

Based on our observations, we hypothesize that infection and subsequent damage of brain vascular pericytes by SARS-CoV-2 and perivascular inflammation may lead to impairment of the BBB, instigating neurological complications and possibly virus entry into the CNS. In line with our report, two recent studies observed vascular leakage and perivascular immune infiltration in the brain of COVID-19 patients, but without the crucial link to ACE2 expression by, and infection of, pericytes [24,25]. However, it is still an outstanding question whether SARS-CoV-2 is overtly neurotropic or if the neurological symptoms associated with COVID-19 are secondary to events related to the systemic host response [26]. Although solely based on the comparable abundance of GFAP (a marker for activated astrocytes) in the tissues, our observations do not provide support for the hypothesis of a cytokine storm. However, increased levels of GFAP have been detected in the plasma of COVID-19 patients [27]. Nevertheless, immune activation markers β2-microglobulin and neopterin were previously found to be elevated in the CSF of COVID-19 patients [28]. In addition, a recent scRNA-seq study on the brains of eight COVID-19 patients revealed an increase in inflammatory genes. More importantly, the observed inflammation of the BBB did not require an active viral infection, possibly explaining our inability to detect SARS-CoV-2 in all COVID-19 cases [29]. Alternative to a cytokine storm, an enhanced inflammatory response could be triggered by metabolic manipulation of mitochondria that are hijacked by the SARS-CoV-2 infection [30]. Hence, further investigations are warranted to fully clarify whether a systemic inflammatory response is associated with neurological manifestations of COVID-19.

Intriguingly, COVID-19 patients with neurological symptoms presented with a reduced concentration of pericyte-derived sPDGFRβ in the CSF. While our mIHC of brain tissue demonstrated a surprisingly variable occurrence of PDGFRβ^+^ perivascular cells, in line with the results from the CSF analysis, the analysis did not support an overall diminished pericyte coverage of the vasculature of COVID-19 patients. A second, and perhaps more likely, explanation for the reduced expression/shedding of PDGFRβ in COVID-19 patients is that SARS-CoV-2 infection of pericytes diverted the protein synthesis machinery to produce viral proteins, leading to loss of endogenous marker expression [31] and consequential functional impairment. 

An improved understanding of SARS-CoV-2 neurotropism is urgently needed to guide the clinical management of acute neurological symptoms, as well as to define strategies to prevent postinfectious neurological complications. We propose that a possible entry site of SARS-CoV-2 into the CNS goes through ACE2-expressing pericytes. Interestingly, although overt endothelial cell infection by SARS-CoV-2 does not appear to occur [15], a recent investigation determined that radiolabeled Appendix A spike viral protein could be retained on the abluminal side of endothelial cells where it is associated with the capillary glycocalyx in mice or further sequestered by the endothelium [32]. It is thus tempting to speculate that this represents one plausible way to expose pericytes to the viral infection. Furthermore, the absence of brain pericytes in mice results in a disrupted BBB associated with widespread loss of integrity [33]. Conversely, sealing of the BBB following thrombolysis after ischemic stroke has been achieved in clinical trials by treatment with the tyrosine kinase inhibitor imatinib [34,35]. Whether similar interventions aiming to support the integrity of the BBB would alleviate neurological symptoms in COVID-19 patients warrants further studies.

## 4. Materials and Methods

### 4.1. Patients

Excessive brain tissues sampled from six COVID-19 autopsies and seven non-COVID-19 cases were used to create formalin-fixed paraffin-embedded (FFPE) blocks (Appendix A). The use of these samples was approved by the Central Ethical Review Authority in Sweden (2020-02369, 2020-06582, and 2020-01771). Clinical data with details of neurologic symptoms or other signs of brain affection were sought in the referral documents or else in the Regional Medical Records database Melior, which was used also for the diagnostic work-up.

CSF from eight patients with neurological manifestations admitted to the Sahlgrenska University Hospital in Gothenburg, Sweden, was included (Appendix A). Infection with SARS-CoV-2 was confirmed via RT-PCR analysis. Age- and sex-matched non-COVID-19 controls were selected, consisting of patients who were examined because of clinical suspicion of neurological disease, but where no neurochemical evidence was found, based on clinical reference intervals. The use of these samples has been approved by the Regional Ethical Committee in Gothenburg.

### 4.2. Bioinformatics Data Access and Analysis

Expression of *Ace2* was investigated in publicly available scRNA-seq SMART-Seq2 libraries on FACS-sorted non-myeloid brain cells of seven mice (*Tabula Muris*) [16] and in a database of murine vasculature [17,18]. 

Mouse whole brain and hippocampus SMART-seq data (gene expression aggregated per cluster, calculated as trimmed means) from the Allen Brain Atlas consortium was downloaded on 14 October 2020 [36,37]. For expression of Pvalb and Sst neurons, the average was calculated of 13 and 40 cell clusters, respectively. 

Human ACE2 protein expression images were retrieved from the Human Protein Atlas initiative (Version 20.0) [19].

### 4.3. Immunohistochemistry (IHC)

Five-micrometer-thick FFPE tissue sections were dewaxed and rehydrated through xylene and water-based ethanol solutions. Heat-induced epitope retrieval was performed with a pressure cooker (2100 Antigen Retriever, BioVendor, Brno, Czech Republic) in citrate or Tris-EDTA buffer (Agilent Dako, Santa Clara, CA, USA). Following endogenous peroxidase quenching (BLOXALL, Vector Laboratories, Burlingame, CA, USA), tissues were incubated with CAS-block (Thermo Fisher Scientific, Waltham, MA, USA) for 1 h at room temperature (RT) and Ultra V block (Thermo Fisher Scientific, Waltham, MA, USA) for 5 min. Primary antibodies (Appendix A) diluted in CAS-block were applied for 30 min, followed by UltraVision ONE HRP polymer (Thermo Fisher Scientific, Waltham, MA, USA) for 30 min, at RT. The ImmPACT DAB substrate (Vector Laboratories, Burlingame, CA, USA) was applied. Tissues were counterstained with hematoxylin, dehydrated, and mounted with Cytoseal 60 (Thermo Fisher Scientific, Waltham, MA, USA). Imaging was performed with an automated BX63 microscope connected to a DP-80 camera (Olympus, Tokyo, Japan).

### 4.4. Multiplexed IHC (mIHC)

FFPE sections used for IHC were subjected to multiplexed labeling following optimized protocols established in the lab. All materials were from Akoya Biosciences (USA), including the Vectra Polaris scanner for imaging and the PhenoChart/InForm software. Following slide preparation, sections underwent staining cycles (Appendix A)—including blocking, primary antibody incubation, HRP tagging, and labeling with OPAL-conjugated tyramide substrate—and a stripping procedure to remove unbound primary antibody/HRP. A counterstain with DAPI preceded the mounting with ProLong Diamond antifade (Thermo Fisher Scientific, Waltham, MA, USA). 

The composite images were generated by removing inherent autofluorescence signal from an unstained section, as well as by comparing fluorescence intensities to those of a spectral library.

### 4.5. Soluble PDGFRβ ELISA 

sPDGFRβ concentration in the CSF was measured by sandwich ELISA (Thermo Fisher Scientific, Waltham, MA, USA), as previously described [38]. Statistical Mann–Whitney U-test was performed using Prism (GraphPad Software, San Diego, CA, USA). The significance level was set at *p* < 0.05, two-sided.

## Figures and Tables

**Figure 1 ijms-22-11622-f001:**
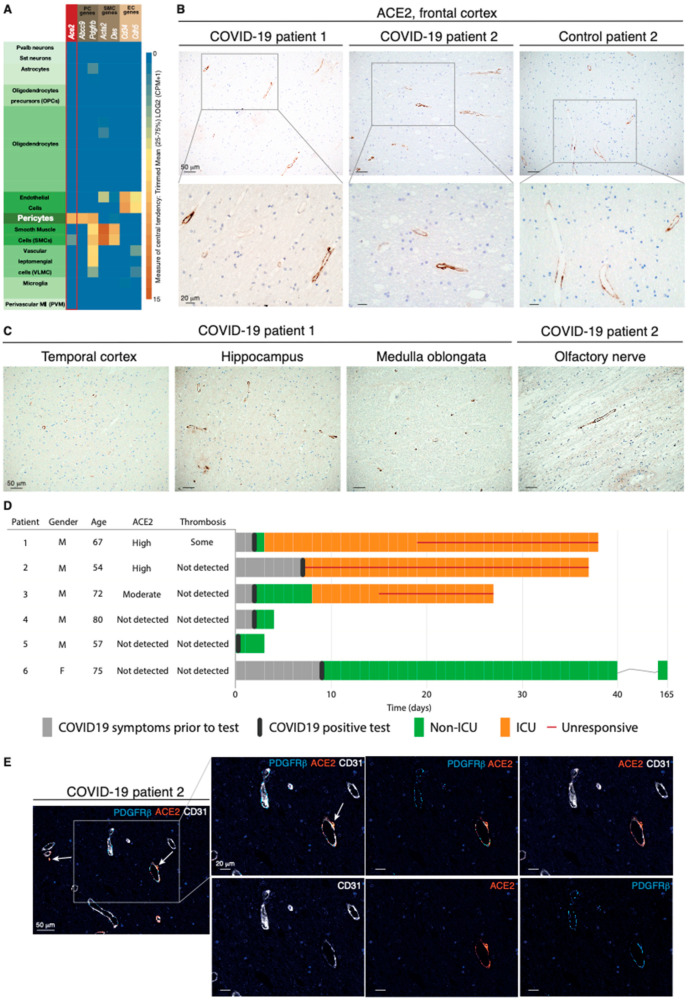
The ACE2 receptor is expressed by pericytes in murine and human brains. (**A**) Expression of *Ace2* in cell types in the mouse brain; cell types are annotated based on the Allen Mouse Brain Atlas. (**B**) Representative IHC staining of perivascular ACE2 in the frontal cortex of two COVID-19 patients and one control individual. Cell nuclei are counterstained with hematoxylin (blue). (**C**) Representative IHC staining of perivascular ACE2 in different brain regions of the two COVID-19 patients in subfigure B. Cell nuclei are counterstained with hematoxylin (blue). (**D**) Clinical course of the COVID-19 patients included in the study. For each patient, appearance of symptoms, hospitalization, infection status, and progression to death are included, together with the *postmortem* evaluation of ACE2 immunoreactivity and thrombosis in the CNS. (**E**) Fourplex mIHC staining of the frontal cortex of a COVID-19 patient. The composite image depicts CD31 (endothelial cells, white), PDGFRβ (pericytes, cyan), and ACE2 (orange). Cell nuclei are counterstained with DAPI (blue). The white arrows indicate ACE2-positive signal in the abluminal side of CD31. The intensity of each individual OPAL fluorophore and the combined PDGFRβ/ACE2 and CD31/ACE2 overlays are presented in individual photomicrographs.

**Figure 2 ijms-22-11622-f002:**
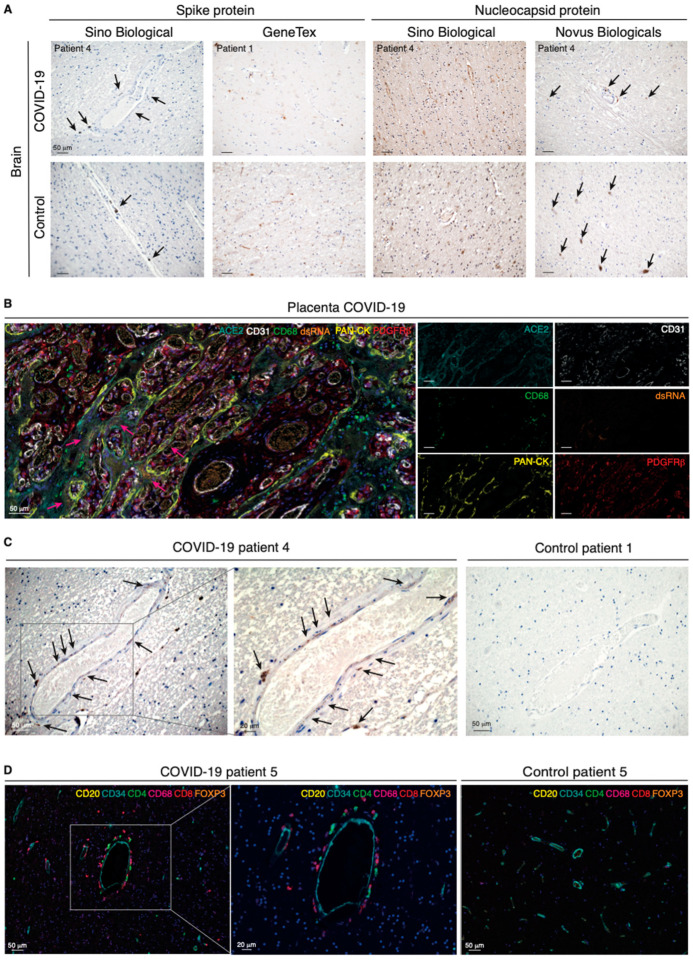
Perivascular infection by SARS-CoV-2 is paralleled by perivascular inflammation in the brain of COVID-19 patients. (**A**) Immunohistochemical detection of viral components in COVID-19-infected patients and non-COVID-19 controls. Cell nuclei are counterstained with hematoxylin (blue). The black arrows indicate the chromogenic deposition of the 3,3’-diaminobenzidine (DAB) substrate. (**B**) Representative field of a 7-plex mIHC staining panel of placental tissue infected with SARS-CoV-2. The magenta arrows indicate accumulation of viral dsRNA in correspondence of the ACE2-positive areas by the specialized epithelial layer of syncytiotrophoblast in the placenta. The intensity of each OPAL fluorophore is further presented in individual photomicrographs. (**C**) Immunohistochemical detection of dsRNA in the cerebral cortex of a COVID-19 patient and in a non-COVID-19 control. Cell nuclei are counterstained with hematoxylin (blue). Black arrows indicate deposition of the DAB substrate. (**D**) Composite mIHC image of the perivascular immune cell infiltration in the frontal cortex of a COVID-19 patient and in a control individual. The antibody panel was designed for the concomitant detection of CD34 (endothelium) and five immune cell markers: CD4 (T helper cells), CD8 (cytotoxic T lymphocytes), CD20 (B cells), CD68 (macrophages), and FOXP3 (regulatory T cells).

**Figure 3 ijms-22-11622-f003:**
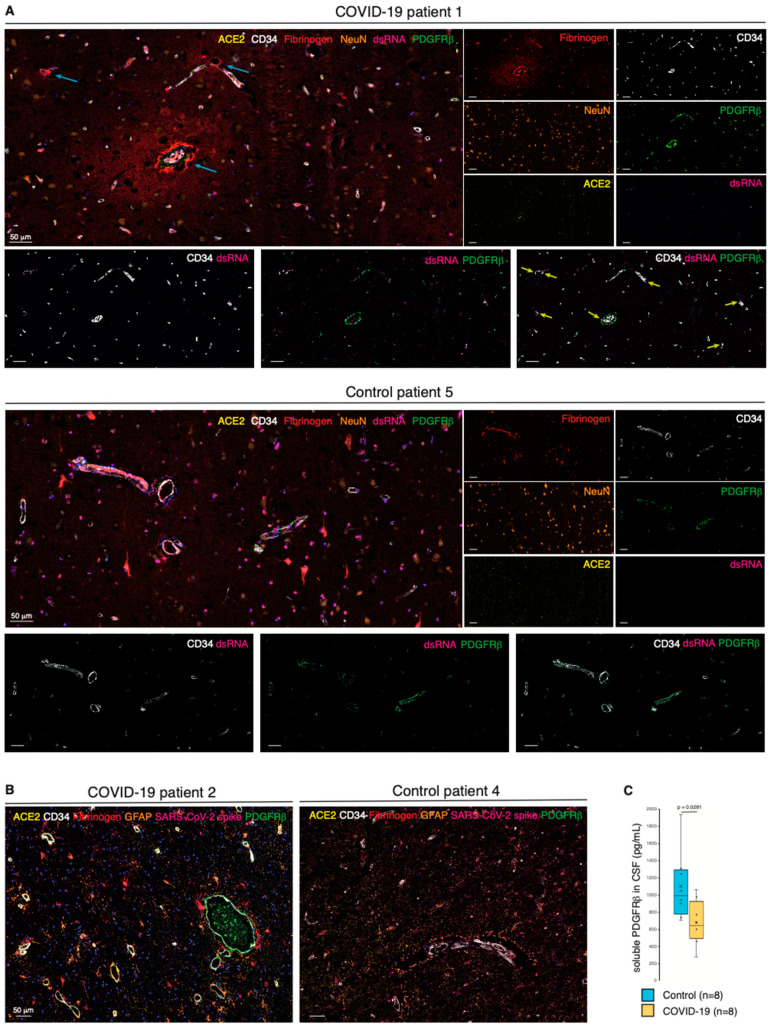
Pericyte infection leads to vascular leakage in the CNS. (**A**) Composite mIHC of the frontal cortex of a COVID-19 patient and a control individual. The fields highlight the fibrinogen halo surrounding leaky blood vessels following SARS-CoV-2 infection. The images depict the neurovascular unit (CD34, PDGFRβ, and ACE2), fibrinogen, viral dsRNA, and neurons. The intensity of each OPAL fluorophore is further presented in individual photomicrographs. The cyan arrows indicate fibrinogen leakage; yellow arrows highlight points of converging PDGFRβ/dsRNA staining. (**B**) Composite mIHC of the frontal cortex of a COVID-19 patient and a control individual. The fields focus on astrocyte priming as a readout of local neuroinflammation. The images depict the neurovascular unit (CD34, PDGFRβ, and ACE2), fibrinogen, SARS-CoV-2 spike protein, and astrocytes (GFAP). (**C**) Boxplot of the concentration of soluble PDGFRβ (pg/mL) in the CSF of COVID-19 patients and non-COVID-19 controls (circles: individual measurements, cross: cohort average).

## Data Availability

The authors confirm that the data supporting the findings of this study are available within the article and its Appendix A.

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
