# Peer review of "Infection of Brain Pericytes Underlying Neuropathology of COVID-19 Patients"

_ijms, 2021, doi:10.3390/ijms222111622_

Round 1

Reviewer 1 Report

Comments to the authors:

 The Manuscript "Infection of brain pericytes underlying neuropathology of COVID-19 patients" by Matteo Bocci et al; with an interesting subject because after Covid 19 people are developing neurological problems. The manuscript has been written in a clear and orderly way and the cited references are very updated and generally appropriate to support the sentences. However, I have some comments herewith to incorporate which will make the manuscript more appropriate to the researchers and readers. Here follows a detailed list of it

Comments:

1- Please justify to use of a few samples i.e. six COVID-19 autopsies and 7 non-COVID-19 for this study?

2-Brain pericytes are perivascular cells that regulate capillary function, and this localization puts them in a pivotal position for the regulation of the central nervous system (CNS) and inflammatory responses at the neurovascular unit after Covid 19, so I would request to measure Inflammatory markers, which is very common to these patients.

3-Please discuss pathways shift during covid 19. Please see below;
Mitochondrial Modulations, Autophagy Pathways Shifts in Viral Infections: Consequences of COVID-19. Int. J. Mol. Sci. 2021, 22(15), 8180.

Author Response

Please see attachment for point-by-point responses to reviewer's comments.

Reviewer 2 Report

In this research article by Bocci et al, the authors attempt to demonstrate the high expression of ACE2 in brain pericytes and associated neurological symptomology due to perivascular inflammation as well as a disrupted blood-brain barrier (BBB). This research article is well designed and highly informative in area of Covid-19. There are satisfied hypotheses and speculation reasonable in the light of the literature. However, there are some minor suggestions to improve the purpose of this review article. For instance

  1. The authors have reported about the expression of ACE2 in different regions of brain in line 70. However, they did not mention about astrocytes which is an important component of BBB and need to be covered.
  2. Line 112, ‘for PDGFRβ, while others expressed both markers’ does not make any sense. The authors should re-write the sentence.
  3. I would suggest writing ‘RT-PCR’ instead of rtPCR in line 261.
  4. The authors should include a list of abbreviation at the end of the manuscript.

Author Response

(The authors gave the same response as above.)
